# The Reaction Behavior of 2CaO·SiO_2_ with CaO–SiO_2_–FeO–P_2_O_5_ Slag

**DOI:** 10.3390/ma15196594

**Published:** 2022-09-22

**Authors:** Yansong Song, Xiaojun Hu, Kuochih Chou

**Affiliations:** State Key Laboratory of Advanced Metallurgy, University of Science and Technology Beijing, Beijing 100083, China

**Keywords:** dephosphorization, 2CaO·SiO_2_, solid solution, slag

## Abstract

It is important to clarify the reaction behavior of 2CaO·SiO_2_ (C_2_S) during hot metal dephosphorization. In this study, C_2_S was prepared and added to steel slag to investigate the reaction of C_2_S particles with CaO–SiO_2_–FeO–P_2_O_5_ slag at 1723 K. The diffusion coefficient of phosphorus in C_2_S was calculated. In addition, the influence of the addition of BaO to C_2_S was discussed. The results show that the diffusion coefficient of phosphorus in C_2_S is 9.23 × 10^−14^ m^2^·s^−1^. The Ca in C_2_S can be replaced by Ba. Small particles in the solid solution were easily generated from the C_2_S body by the addition of BaO, which is beneficial for improving the phosphorus partition between the C_2_S solid phase and the liquid phase of the slag.

## 1. Introduction

During dephosphorization, the state of CaO-based fluxes differs at various stages of metal treatment and strongly depends on its chemical composition. In particular, solid phases with high melting points usually cause a series of operational problems, such as increases in slag volume, lime consumption, and difficulty in slag recycling. To mitigate these problems, a new double slag converter steelmaking process was proposed [1]. In this process, the decarburization slag from the upper furnace was reused in the next furnace for dephosphorization and desiliconization. P_2_O_5_ is a part of 3CaO·P_2_O_5_, and 3CaO·P_2_O_5_ can react with C_2_S to form a 2CaO·SiO_2_–3CaO·P_2_O_5_ (C_2_S–C_3_P) solid solution [2,3,4,5,6,7]. However, the reaction behavior of 2CaO·SiO_2_ in the dephosphorization process needs further clarification.

Several studies on the dephosphorization behavior of multiphase slag have been conducted. These studies mainly focused on the reaction mechanism of solid CaO with steel slag to form a C_2_S–C_3_P solid solution, mass transfer behavior of phosphorus from the liquid slag phase to the C_2_S solid phase, phosphorus partition between the C_2_S particles and steel slag, isothermal crystallization and crystallization kinetics of C_2_S–C_3_P solid solutions, the formation free energies of C_2_S–C_3_P solid solution, and activities of the components in C_2_S–C_3_P solid solutions [7,8,9,10,11,12,13,14,15,16,17]. Meanwhile, some studies have suggested that the phosphorus partition between the C_2_S solid solution phase and liquid phase is related to the structure and density of the solid solution phase. Xie et al. [18] studied the structure and density of solid solutions with various P_2_O_5_ contents and demonstrated that the structure of the C_2_S–C_3_P solid solution became more compact and the density increased with increasing P_2_O_5_ content. The density of the C_2_S particles varied along their radii in proportion to the phosphorus content, thus causing residual stresses. During solid solution dephosphorization, high residual stress was formed between the edge and interior of the solid solution, resulting in the edge of the solid solution easily falling off from the solid solution body. Thus, the dephosphorization efficiency of the C_2_S solid solution was improved. Therefore, it is important to study the structure and density of C_2_S–C_3_P solid solutions. Since BaO is more alkaline than CaO and is also a good flux, BaO may have a significant impact on dephosphorization. However, the dephosphorization efficacy, structure, and density of 2CaO·SiO_2_–3CaO·P_2_O_5_ solid solutions with BaO have not been studied.

In this study, artificially prepared C_2_S was added to steel slag. By studying the reaction of C_2_S particles with CaO–SiO_2_–FeO–P_2_O_5_ at 1723 K, the diffusion coefficient of phosphorus in C_2_S could be obtained. The generated C_2_S–C_3_P solid solution and peeling phenomena were analyzed by optical microscopy and scanning electron microscopy (SEM) combined with energy dispersive X-ray spectroscopy (EDS). The effect of the addition of BaO to the C_2_S–C_3_P solid solution on the density and peeling of C_2_S was studied.

## 2. Experimental

### 2.1. Sample Preparation

FeO was prepared by mixing electrolytic iron powder and dried reagent-grade Fe_3_O_4_ with a molar ratio of 1:1, pressing it into a cylindrical shape, placing it in a pure iron crucible, and holding for 12 h under CO and CO_2_ atmospheres at a 1:1 flow rate ratio and 1373 K temperature. Subsequently, the crucible was quickly removed and cooled via purging with argon. The sample was ground to a powder for X-ray diffraction (XRD) analysis, and the results are shown in Figure 1. It can be seen that the sample was pure FeO.

To prepare the C_2_S, reagent-grade CaCO_3_ was first heated to 1273 K for 10 h and decomposed to obtain CaO, then mixed with SiO_2_ at a molar ratio of 2:1. To avoid pulverization, B_2_O_3_ was added to the sample, such that the final mixture was 1–2% B_2_O_3_. The mixture was pressed into a cylindrical shape and heated in an Al_2_O_3_ crucible at 1773 K for 24 h [19]. The sample was then cooled and ground into a powder. The XRD results in Figure 2 show that the sample was in the C_2_S phase.

The prepared CaO, reagent-grade SiO_2_, 3CaO·P_2_O_5_, and synthesized FeO were mixed thoroughly in an agate mortar. The chemical compositions of the mixtures are listed in Table 1.

### 2.2. Experimental Apparatus and Procedure

In this study, an electric resistance furnace with an accuracy of ±3 K was used to heat the slag sample, as shown in Figure 3. A mixture of 20 g of CaO–SiO_2_–FeO–P_2_O_5_ slag and 2 g of solid iron was placed in an Al_2_O_3_ crucible—the solid iron was used to control the oxygen partial pressure based on the Fe/FeO equilibrium in the experiment [15]. The mixture was first heated to 1173 K at a rate of 10 K/min and then further heated to 1723 K at a rate of 5 K/min in an argon atmosphere. To ensure complete melting, the molten slag was held at 1723 K for 0.5 h. Subsequently, 10 g of C_2_S powder (120–250 µm diameter) was added to the crucible while recording the time and stirring quickly using a thin alumina stick. An alumina stick was used to collect 1 g slag samples at specific time intervals (0, 60, 120, 180, and 240 s), with 0 s being set as the time that the first sample was taken. The outer surface of the dipped slag sample was used to observe the morphology and perform component analysis using SEM/EDS. To reduce the inaccuracy of the EDS analysis, multiple analyses of each EDS point were performed to obtain an average value.

## 3. Results and Discussion

### 3.1. Diffusion of P_2_O_5_ in C_2_S

The morphology of the P_2_O_5_ in the C_2_S particles is shown in Figure 4. EDS point measurements of the erosion layer were taken from the edge to the interior of the C_2_S to obtain its composition. The white diffusion layer could be clearly identified. The slag eroded inward from the edge of the particles after the addition of C_2_S. Some small particles, in which the main component was n·C_2_S–C_3_P, were observed in the diffusion layer. Furthermore, some small particles were scattered in the slag because of the initial stirring. The chemical compositions of the different positions marked in Figure 4 are shown in Figure 5. The P_2_O_5_ content of the C_2_S particles showed a decreasing trend from the edge toward the center. The FeO adjacent to the n·C_2_S–C_3_P solid melt was diffused into the C_2_S particles.

The diffusion coefficient of phosphorus through the slag phase was obtained from Equation (1) by Shen et al. [20].
(1)$$\ln(\omega /\omega_{0})=\frac{D}{\delta }\cdot \frac{A}{V}\cdot t$$

ω—P_2_O_5_ mass fraction in C_2_S;

ω_0_—Initial P_2_O_5_ mass fraction in C_2_S;

D—The diffusion coefficient of P_2_O_5_ in C_2_S, m^2^·s^−1^;

δ—Thickness of generated solid solution layer, m;

A—Surface area of reaction interface, m^2^;

V—The volume of product solid solution layer, m^3^;

t—Reaction time, s.

The average P_2_O_5_ content (disregarding measurements that were below 1% P_2_O_5_ by mass) of the solid solution layer of the product was used in Equation (1). The relationship between ln(ω/ω_0_) and time is shown in Figure 6. We fitted the data points using a regression analysis method. The slope of the fitted line was 9.23 × 10^−4^. It was assumed that δ and A/V were 1 × 10^−5^ m and 1 × 10^5^ m^−1^, respectively. Finally, Equation (1) was used to obtain the diffusion coefficient D = 9.23 × 10^−14^ m^2^·s^−1^. Dou et al. [21] measured the diffusion coefficients of P_2_O_5_ in 40.83%CaO–29.17%SiO_2_–20.00%Fe_t_O–10.00%P_2_O_5_ slag at 1623 K and 1673 K to be 2.70 × 10^−14^ m^2^·s^−1^ and 2.98 × 10^−14^ m^2^·s^−1^, respectively. Shen et al. [20] measured the diffusion coefficient of P_2_O_5_ in 40.9%CaO–34.1%SiO_2_–20.0%FeO–5.0%P_2_O_5_ slag at 1773 K to be 5.00 × 10^−13^ m^2^·s^−1^. The diffusion coefficient of phosphorus was determined by the composition of the slag and reaction temperature. The diffusion coefficient of P_2_O_5_ increased with increasing temperature. The calculated diffusion coefficient (D) was between those calculated by Dou et al. [21] and Shen et al. [20] because the reaction temperature used in this study was between those of Dou et al. [21] and Shen et al. [20].

The relationship between the P_2_O_5_ and FeO content is shown in Figure 7. The P_2_O_5_ and FeO contents were fitted using Gaussian line shapes. There was a positive correlation between the P_2_O_5_ and FeO contents in the 2CaO·SiO_2_ solid melt. At present, it is known that the dephosphorization product of the CaO–SiO_2_–FeO_t_–P_2_O_5_ slag system existed in the form of an n·C_2_S–C_3_P solid melt [2,3,4,5,6,7]; however, the formation process of the n·C_2_S–C_3_P solid melt was not sufficiently clear. Dou [21] suggested that C_3_P in the slag could react with C_2_S to form an n·C_2_S–C_3_P solid melt. However, Su [22] demonstrated through molecular ion theory calculations that phosphorus in the slag should exist in the form of 3FeO·P_2_O_5_. Figure 7 shows that the P_2_O_5_ content in the solid increased with an increase in FeO. The phosphorus in the slag may have been in the form of 3FeO·P_2_O_5_. Furthermore, FeO remained in the particle after the formation of the n·C_2_S–C_3_P solid melt. Therefore, the P_2_O_5_ and FeO contents of the C_2_S solid melt were positively correlated. 

### 3.2. Clarification of Generated Solid Solution in Rim of C_2_S

According to Inoue [23], the phosphorous transfer rate from the slag to C_2_S particles with diameters between 20 and 50 µm was considerably fast, and the transformation from the C_2_S particle into the C_2_S–C_3_P particle occurred within 5 s. When the particles existed in clusters, only the outer surface of the cluster (to a depth of 5 µm) reacted to form a C_2_S–C_3_P solid solution within 5 s. When the C_2_S particles were large, the generated diffusion layer became denser with the formation of the solid product layer, increasing the diffusion resistance of the slag to the unreacted C_2_S particles. Thus, the continuation of the reaction of phosphorus diffusion to C_2_S particles (C_2_S with P_2_O_5_) was limited. When the C_2_S particles were small, the diffusion of phosphorus into the C_2_S particles was rapid.

The solid solution layer of the product was analyzed using optical microscopy and SEM/EDS, as shown in Figure 8 and Figure 9 and Table 2. Figure 8a,b shows that the surfaces of the generated solid solution layer and small particles that were shed were irregular and showed some gully shapes, thereby leading to a height difference on the particles’ surfaces, as shown in Figure 8e. Although the largest height difference was 0.4 µm and particle sizes were relatively small, a trend could still be observed in Figure 9 and Table 2, where the phosphorus content of the upper gully shapes was lower than that on the bottom. The size of the SEM’s electron beam area decreased the measurement accuracy, but the analysis could be used as a basis for qualitative analysis. To reduce the inaccuracy of EDS, multiple analyses of each dot were performed, eventually obtaining the average value. Therefore, the EDS data were still reliable. According to the results of the research by Xie et al. [18], a solid solution with a high phosphorus content has a high density. The residual stress was generated in a solid particle when the density was locally inhomogeneous owing to composition fluctuation. The high phosphorus content of the C_2_S–C_3_P solid solution led to the surface gradually peeling off owing to residual stress. Thus, gully shapes were formed and the C_2_S–C_3_P solid solution particles peeled off. 

### 3.3. Influence of Addition of BaO in C_2_S

Some studies have suggested that the phosphorus partition between the C_2_S solid solution phase and the liquid phase is related to the structure and density of the solid solution phase. Xie et al. [18] studied the structure and density of a solid solution with varying P_2_O_5_ contents and reported that the structure of the C_2_S–C_3_P solid solution became more compact and the density increased with increasing P_2_O_5_ content. During solid solution dephosphorization, higher residual stress is formed between the edge and interior of the solid solution because of density differences, and the edge of the solid solution easily falls off from the solid solution body. Thus, the dephosphorization efficiency of the C_2_S solid solution is improved. Therefore, it is very important to study the structure and density of C_2_S–C_3_P solid solutions. However, a careful examination of the literature data indicated that the structure and density of the 2CaO·SiO_2_–3CaO·P_2_O_5_ solid solution with varying BaO contents remain to be elucidated. Therefore, the density of the C_2_S–C_3_P solid solution with barium was determined. The composition of the solid solutions is listed in Table 3. To prepare solid solutions with different BaO contents, the pure chemical reagents CaO, SiO_2_, 3CaO·P_2_O_5_, and BaCO_3_ were first weighed in proportion, mixed thoroughly, heated at 1673 K for 10 h, and then cooled in air to obtain solid solutions. Part of the sample was ground into a powder with a suitable particle size for testing the sample density and XRD analysis. Another part of the sample was mosaicked and polished for SEM inspection to characterize the surface morphology.

The experimental samples were analyzed by XRD (Smart (RIGAKU), Tokyo, Japan) with Cu-K radiation at 40 KV and 30 mA over the range of 2θ between 20° and 60°, and the results are shown in Figure 10. With the addition of BaO, the intensity of the CaO XRD diffraction peaks was enhanced, which may be attributed to the substitution of CaO with BaO in C_2_S–C_3_P. At present, there are no standard XRD profiles of C_2_S–C_3_P at varying phosphorus contents, while the structure of the solid solution containing 20% P_2_O_5_ was the same as that of α-C_2_S (JCPDS NO. 86-0401) [24]. The intensity of most of the α-C_2_S XRD peaks increased with the addition of BaO. Additionally, the peaks shifted slightly to the left, indicating that the lattice constant and the crystal plane spacing increased with increasing BaO content. When the BaO contents in the solid solution were 5%, 10%, and 15%, the deviations were approximately 0.06°, 0.19°, and 0.31°, respectively, compared with that when the BaO content was 0%, indicating that the barium ion replaced the calcium ion and enlarged the lattice of the solid solution. Since the atomic weight (137.3) of barium was much larger than that of calcium (40.08), the addition of BaO was beneficial for increasing the solid solution density.

The aggregation state of the substances was determined using SEM. Xie et al. [18] observed that the solid solution became more compact with increasing P_2_O_5_ content. The SEM images of C_2_S–C_3_P with varying BaO contents are shown in Figure 11. It can be seen that the formed solid solution was more compact with the addition of BaO. When BaO was present in the steel slag, the compactness of the artificially prepared C_2_S solid solution gradually increased from the edge to the interior of C_2_S with different components in C_2_S. This increased the residual stress at the edge of the C_2_S solid solution, making it easier for the edge of the solid solution to fall off during dephosphorization, which was beneficial for improving the dephosphorization rate of C_2_S. 

The experimental samples were further analyzed using a true density analyzer (Ultra PYC 1200e, Knoxville, TN, USA), and the results are shown in Figure 12. For comparison, the experimental data obtained by Xie et al. [18] are also shown in Figure 12. In this study, when the BaO content was 0 and the P_2_O_5_ content was 20%, the density of the sample was consistent with that reported by Xie et al. As the BaO content increased from 0 to 15%, the true density of the solid solution increased linearly from 3.12 to 3.36 g/cm^3^. This indicates that the variations in the structure and compaction of the solid solution eventually led to an increase in the solid solution density. The number of electrons outside the barium and calcium atoms was the same, while the barium ionic radius (0.134 nm) was larger than the calcium ionic radius (0.099 nm), which enlarged the crystal of the solid solution, and the deviations were mentioned in the XRD analysis. Since the atomic weight of barium (137.3) is larger than that of calcium (40.08), the density of the solid solution increased when the BaO content increased from 0% to 15%. This was beneficial for improving the phosphorus partition between the C_2_S solid phase and liquid phase of the steel slag.

Mixtures of 20 g of CaO–SiO_2_–FeO–BaO–P_2_O_5_ slag were placed in an Al_2_O_3_ crucible with an inner diameter of 24 mm and a height of 50 mm. The crucible was then placed in the constant-temperature zone of the electric resistance furnace. Their compositions are listed in Table 4. The mixtures were heated to 1173 K at a rate of 10 K/min, then further heated to 1723 K at a rate of 5 K/min and held at that temperature for 0.5 h to ensure they were fully melted. Subsequently, 10 g of C_2_S powder (120–250 µm diameter) was added to the crucible while stirring quickly using a thin alumina stick and the time was recorded. The crucible was then removed for water quenching after a 60 s reaction time. The slag sample morphology and component composition were determined using optical microscopy and SEM/EDS.

Figure 13 shows optical microscopy images of the C_2_S particles after a reaction time of 60 s. As shown in Figure 13, the C_2_S particles were broken down into many smaller particles with the addition of BaO, indicating that the dephosphorization efficiency of the C_2_S solid solution was improved. Area and point scan analyses by SEM/EDS were conducted and the results are shown in Figure 14 and Table 5, which confirm that Ba has entered the small particles of C_2_S, although its content in C_2_S was low. As previously described in Section 3.2, with the addition of BaO, residual stress was generated in a solid particle when the density was locally inhomogeneous due to composition fluctuation, and the C_2_S–C_3_P solid solution particles were more easily peeled off, as shown in Figure 13. Therefore, based on a comparison with the morphology of C_2_S particles without BaO addition in Figure 4, it can be concluded that the influence of Ba on the decomposition of C_2_S into small particles was significant, which was beneficial for improving the dephosphorization effect of C_2_S.

## 4. Conclusions

C_2_S powder was added to CaO–SiO_2_–FeO–P_2_O_5_ slag at 1723 K and the reaction between them was investigated. The solid solution generated at the surface of C_2_S was characterized, and the influence of the addition of BaO to C_2_S was discussed. The following conclusions were drawn.


1.The phosphorus-rich phase in the slag mainly existed in the form of an nC_2_S–C_3_P solid melt. Phosphorus existed in the form of an nC_2_S–C_3_P solid melt. The diffusion coefficient (D) of phosphorus in C_2_S was 9.23 × 10^−14^ m^2^·s^−1^. There was a positive correlation between the P_2_O_5_ and FeO contents in the C_2_S solid solution.2.The surfaces of the generated solid solution layer and the shed small particles were irregular and showed some gully shapes. The phosphorus content of the raised portion of the gully shape was lower than that at the bottom. The C_2_S–C_3_P solid solution with a high phosphorus content gradually peeled off owing to residual stress.3.The C_2_S solid solution was more compact and its density was improved by the addition of BaO. The Ca in C_2_S could be replaced by Ba. The small particles of the solid solution were more easily generated from the C_2_S body with the addition of BaO, which was beneficial for improving the phosphorus partition between the C_2_S solid phase and the liquid phase of the slag.


## Figures and Tables

**Figure 1 materials-15-06594-f001:**
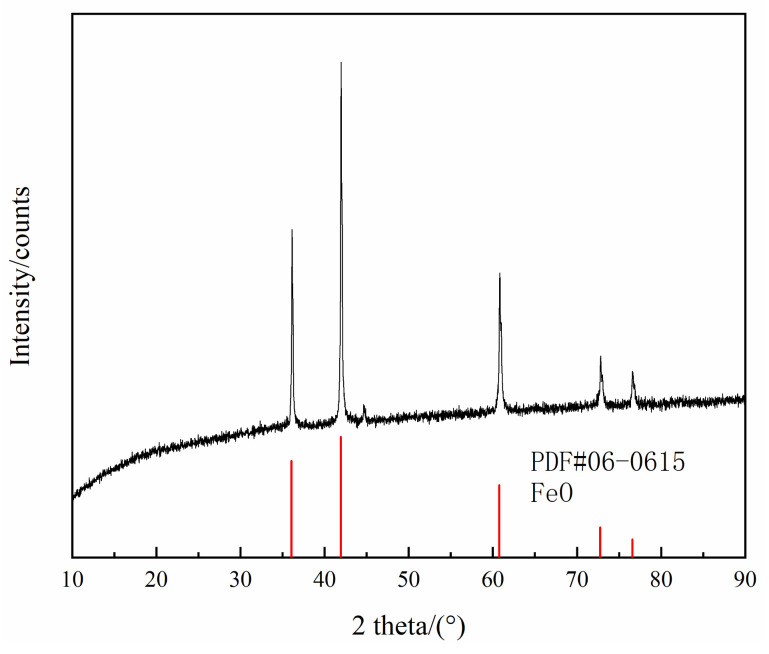
XRD results for FeO powder.

**Figure 2 materials-15-06594-f002:**
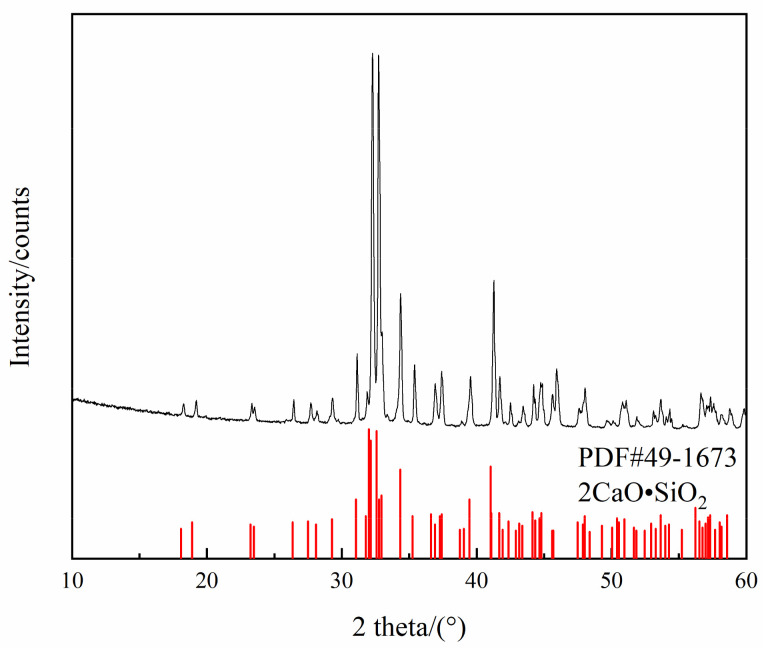
XRD results for C_2_S powder.

**Figure 3 materials-15-06594-f003:**
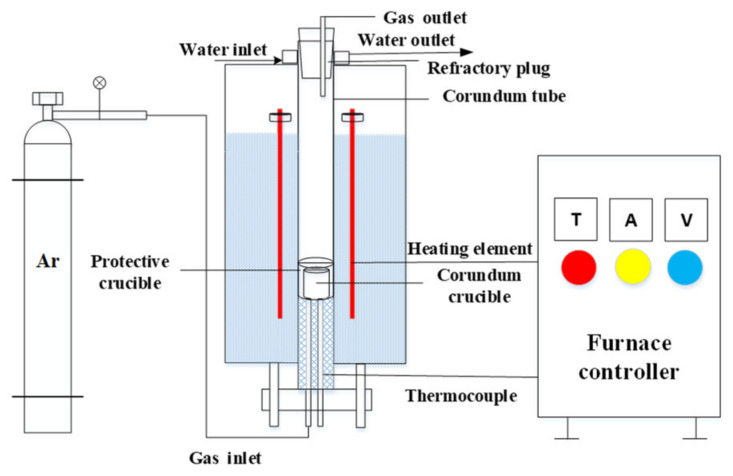
Schematic of the experimental apparatus.

**Figure 4 materials-15-06594-f004:**
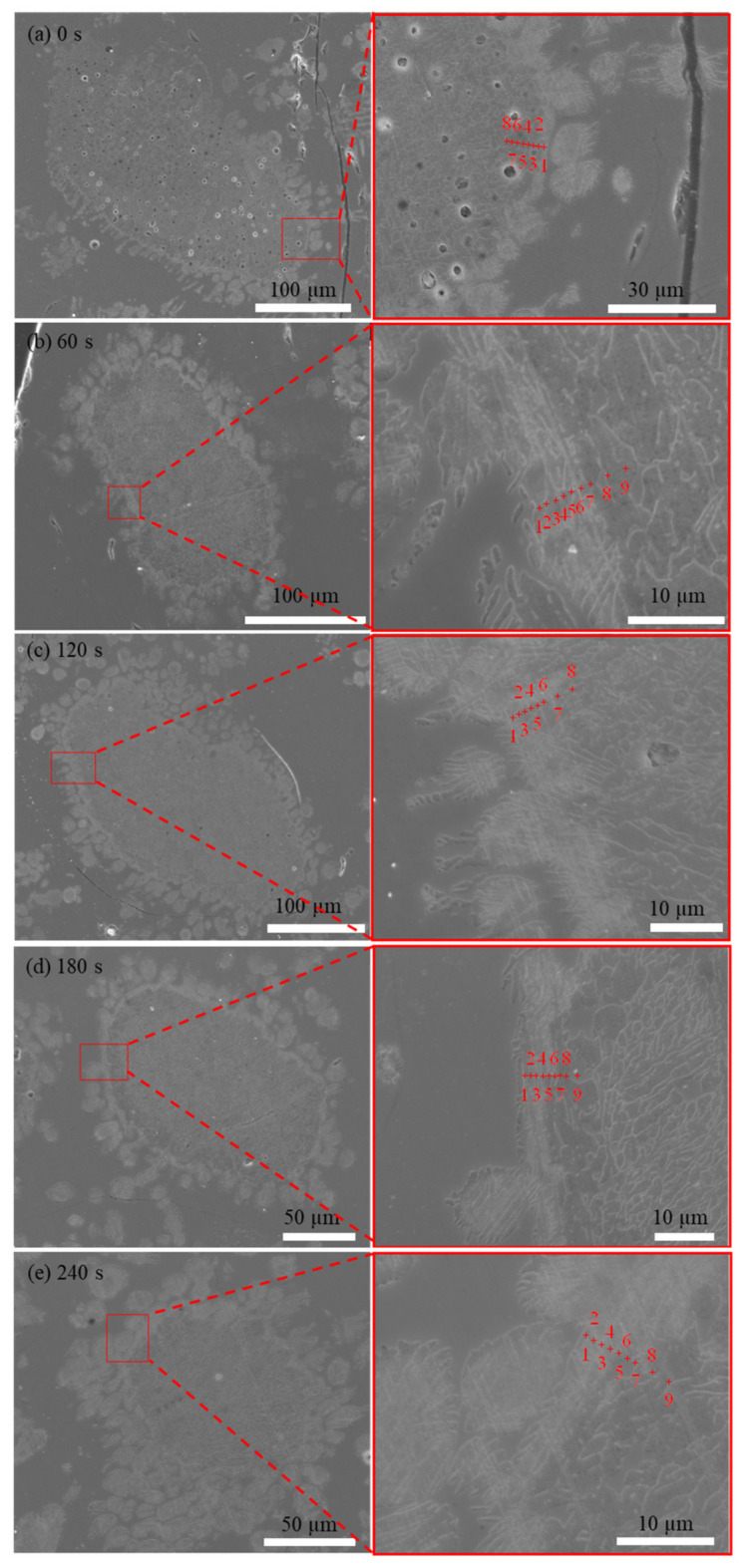
Diffusion morphologies of P_2_O_5_ in the C_2_S particles with different times: (**a**) 0 s; (**b**) 60 s (**c**) 120 s; (**d**) 180 s; (**e**) 240 s.

**Figure 5 materials-15-06594-f005:**
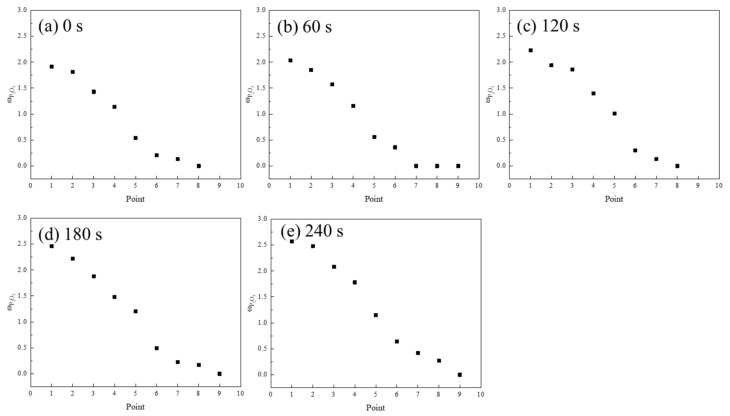
P_2_O_5_ contents in C_2_S particles with different times in Figure 4: (**a**) 0 s; (**b**) 60 s (**c**) 120 s; (**d**) 180 s; (**e**) 240 s.

**Figure 6 materials-15-06594-f006:**
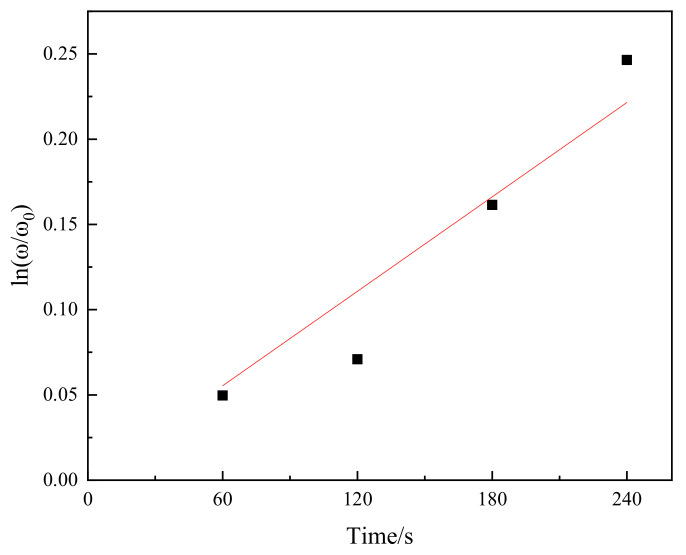
Relationship between $\ln(\omega /\omega_{0})$ and reaction time.

**Figure 7 materials-15-06594-f007:**
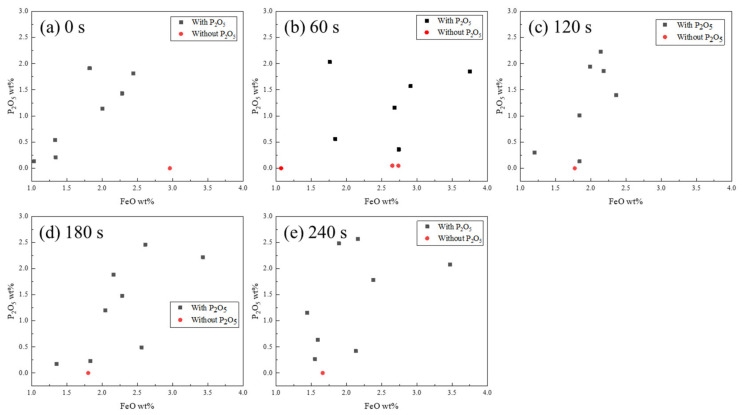
Relationship between the P_2_O_5_ and FeO contents in C_2_S.

**Figure 8 materials-15-06594-f008:**
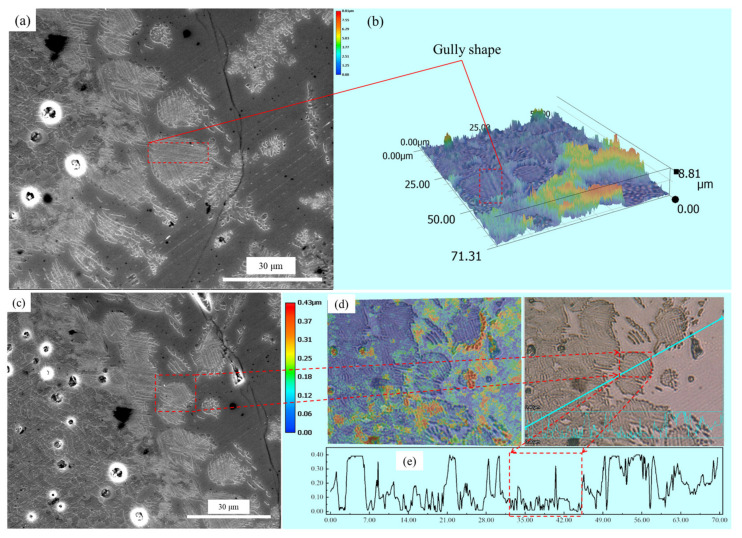
Analysis of the C_2_S–C_3_P solid solution by optical microscopy: (**a**) Visual field 1; (**b**) Surfaces of the generated solid solution layer of visual field 1; (**c**) Visual field 2; (**d**) Surfaces of the generated solid solution layer of visual field 2; (**e**) Height distribution of visual field 2.

**Figure 9 materials-15-06594-f009:**
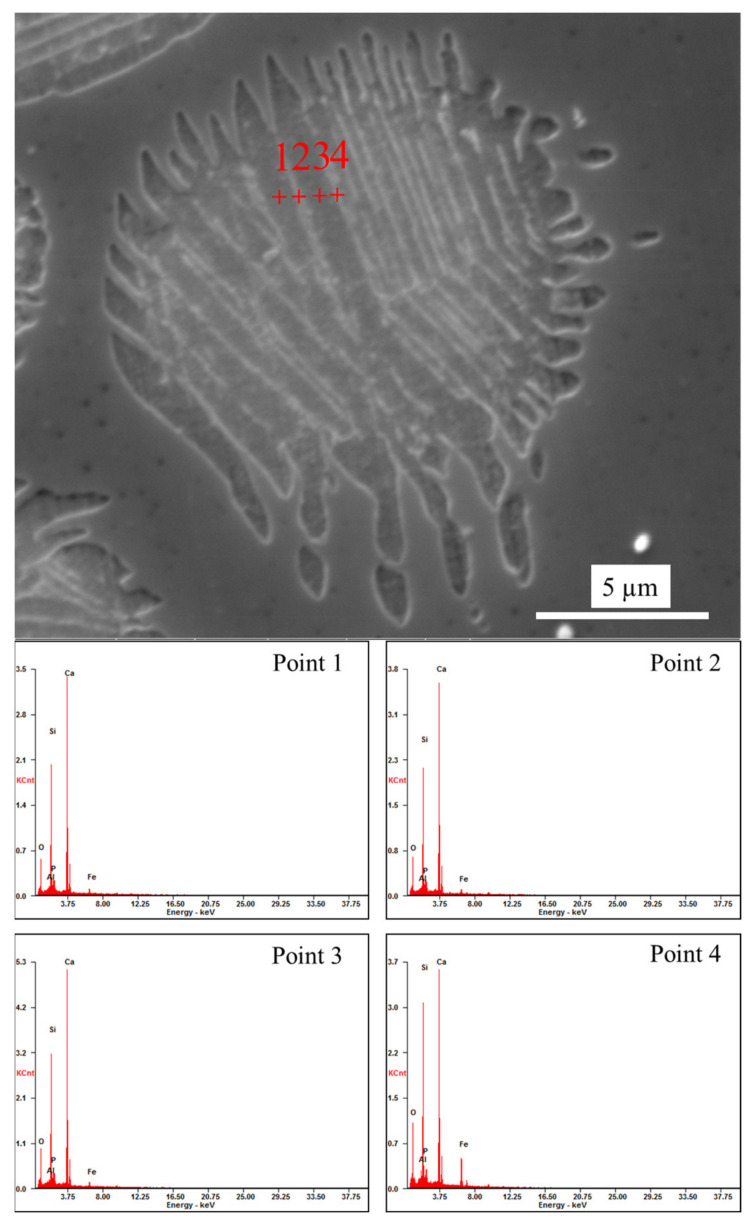
SEM micrograph of a shed small particle.

**Figure 10 materials-15-06594-f010:**
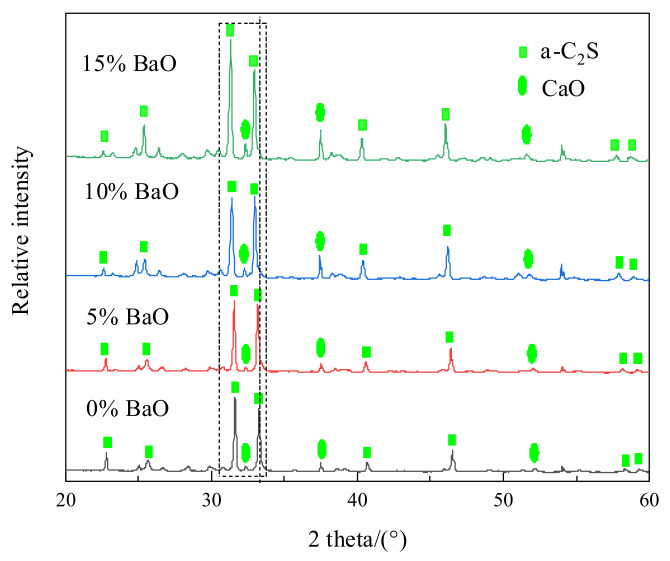
XRD analysis of the C_2_S–C_3_P solid solution with varying BaO content.

**Figure 11 materials-15-06594-f011:**
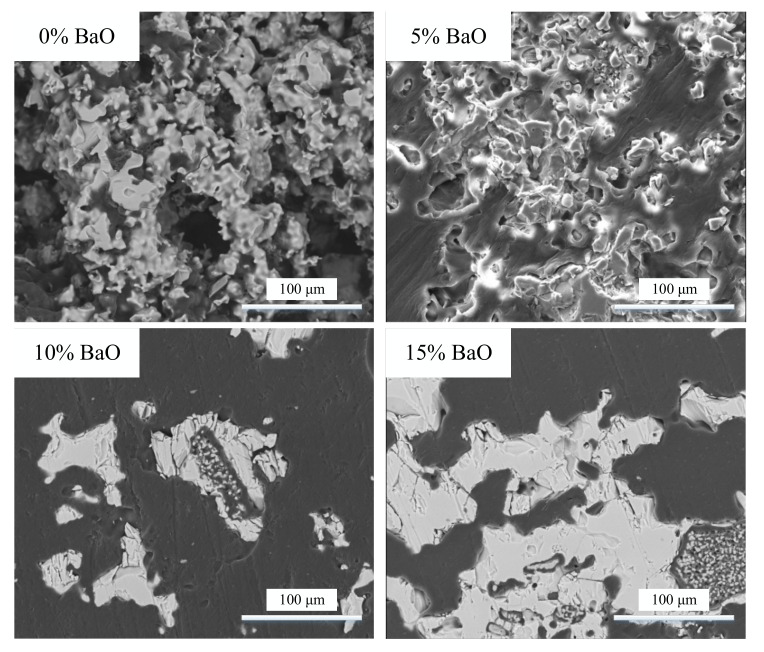
SEM images of C_2_S–C_3_P solid solution under 1000× magnification.

**Figure 12 materials-15-06594-f012:**
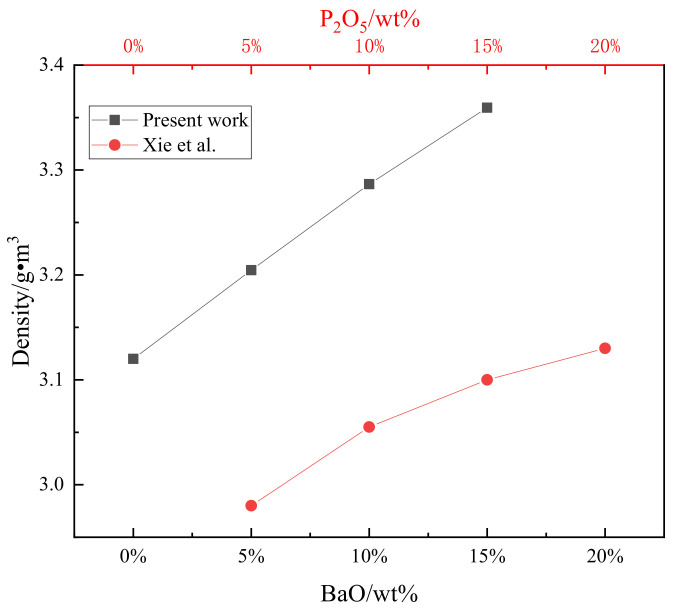
True density of the C_2_S–C_3_P solid solutions with various BaO contents.

**Figure 13 materials-15-06594-f013:**
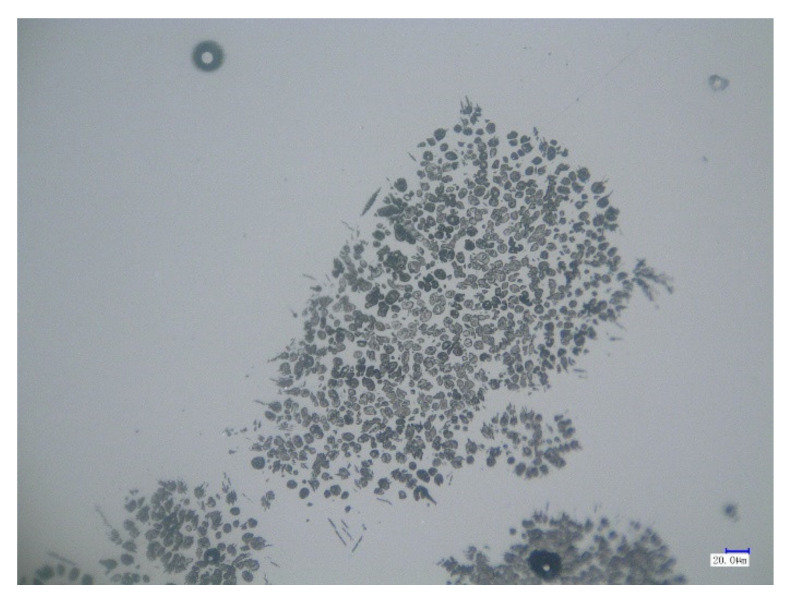
Optical microscopy analysis of C_2_S.

**Figure 14 materials-15-06594-f014:**
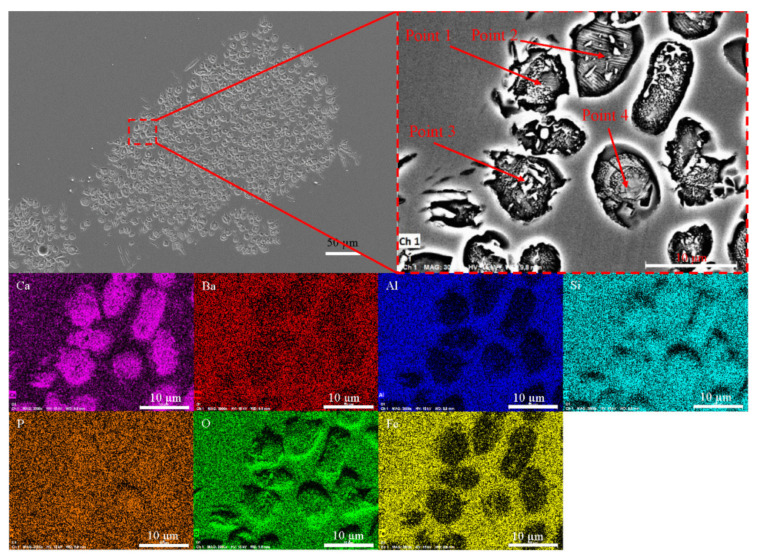
Area and point scan analyses by SEM/EDS.

**Table 1 materials-15-06594-t001:** Chemical compositions of CaO–SiO_2_–FeO–P_2_O_5_ slag (wt.%).

CaO	SiO_2_	FeO_x_	P_2_O_5_	R(CaO/SiO_2_)
41.92	36.08	20.00	2.00	1.16

**Table 2 materials-15-06594-t002:** Element analysis of four points in Figure 9.

Point	Oxygen		Aluminum		Silicon		Phosphorus		Calcium		Ferrum	
	wt.%	at.%	wt.%	at.%	wt.%	at.%	wt.%	at.%	wt.%	at.%	wt.%	at.%
1	23.51	41.16	0.33	0.34	19.21	19.16	1.62	1.46	51.34	35.88	3.99	2.00
2	23.45	41.02	0.07	0.07	19.43	19.36	2.27	2.05	50.93	35.57	3.86	1.93
3	23.47	41.10	0.06	0.07	18.92	18.87	1.96	1.77	52.20	36.49	3.39	1.70
4	24.41	42.29	0.13	0.13	18.97	18.72	2.12	1.90	51.08	35.33	3.30	1.64

**Table 3 materials-15-06594-t003:** Chemical compositions of solid solutions with different BaO content (wt.%).

Sample	CaO	SiO_2_	P_2_O_5_	BaO
1	60.36	19.64	20.00	0
2	57.11	17.89	20.00	5.00
3	53.85	16.15	20.00	10.00
4	50.59	14.41	20.00	15.00

**Table 4 materials-15-06594-t004:** Chemical compositions of CaO–SiO_2_–FeO–BaO–P_2_O_5_ slag (wt.%).

CaO	SiO_2_	FeO_x_	BaO	P_2_O_5_	R(CaO/SiO_2_)
39.20	33.80	20.00	5.00	2.00	1.16

**Table 5 materials-15-06594-t005:** Element analysis of four points in Figure 14.

Point	Oxygen		Aluminum		Silicon		Phosphorus		Calcium		Barium		Ferrum	
	wt.%	at.%	wt.%	at.%	wt.%	at.%	wt.%	at.%	wt.%	at.%	wt.%	at.%	wt.%	at.%
1	39.98	61.15	0.82	0.74	12.40	10.80	1.85	1.46	38.29	23.38	1.79	0.32	4.88	2.14
2	33.72	52.49	0.71	0.66	17.20	15.25	1.91	1.54	47.32	29.41	0.98	0.18	1.06	0.47
3	39.12	59.71	0.65	0.59	12.45	10.82	1.95	1.54	43.91	26.76	1.01	0.18	0.91	0.40
4	35.62	55.14	0.65	0.60	15.75	13.89	2.21	1.77	43.85	27.10	1.21	0.22	2.91	1.29

## Data Availability

Not applicable.

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
