# Peer review of "The Reaction Behavior of 2CaO·SiO2 with CaO–SiO2–FeO–P2O5 Slag"

_materials, 2022, doi:10.3390/ma15196594_

Round 1
Reviewer 1 Report
1. Introduction, Line 27. I would say it is not correct to write, that P2O5 exists in C2S phase with 2CaO SiO2-3CaO P2O5 solid solution. Actually, the high-temperature modifications of dicalcium silicate Ca2SiO4 and tricalcium phosphate Ca3P2O8 form a continuous series of solid solutions at steelmaking temperatures (1500-1900°C). P2O5 is a part of Ca3P2O8 which is the endmember of C2S.
2. Figure 3. The Table 1 below is given in wt.%, the isothermal section in Figure 3 could be presented also in wt.%. Speciality of the calculations with iron is that the boundary conditions always should be mentioned, because the results are completely different if you calculate for example in air (Fe2O3 is stable) or in equilibrium with iron (FeO is stable). The Figure 3 should be recalculated as CaO + SiO2 + FeO + Fe.
3. Figure 10. The table is too small, it is difficult to read.
4. Table 15. If you write in the column, it is Element, you can also write the names of the real elements.
5. 3.3. Influence of Barium. As I see, in your version of FactSage7.0 Ba is also included into thermodynamic description of C2S-C3P. Maybe it could be possible to calculate something with participation of BaO?

Author Response
Thank you very much for reading our paper carefully. And thanks again very much for your concern in our work.

Reviewer 2 Report
The statement that CaO-based fluxes used in a steelmaking process are in a solid-liquid coexisting state is disputable. The state of slag is different at various periods of metal treatment and strongly depends on chemical composition.
The authors should thoroughly reread the text to change meaningless phrases with the wrong causality. After that text should be proofread by a native speaker to remove grammar and style mistakes and sufficiently improve English.
There are a lot of meaningless phrases (especially in the first part of the article):
Lines 24- 25 “NDSP is the decarburisation slag from the upper furnace used in the next furnace for dephosphorisation and desilicification”. The process can not be a slag; secondary “desilicification” is not the correct term –the commonly used term for this subject should be – desiliconization.
Lines 28 – 29 “ The utilisation of C2S-C3P solid solution in the steelmaking slag can greatly reduce the amount of steelingmaking slag and save calcium oxide resources due to the high phosphorus content.”is also senseless - nobody can put C2S-C3P to “save calcium oxide resources due to the high phosphorus content”.
Lines 30-31: Meanwhile, this is what all metallurgical researchers hope to see in the environment of increasingly prominent environmental pollution problems.
Line 45-46: In principle, residual stress is made in a solid particle when density is locally inhomogeneous due to temperature or composition inhomogeneity – strange explanation - inhomogeneous due to … inhomogeneity?
There are many other sentences, so I kindly ask authors to reread their text phrase by phrase to remove unnecessary and make vague statements clear and easy readable.
Statement in lines 83-84: “The chemical compositions of the mixture were listed in Table 1, which was mainly determined based on the ternary phase diagram of CaO-SiO2- FeO system drawn by Factsage 7.0” is not true because components in table 1 do not coincide with predicted by the diagram.
Considering that this diagram is not further used or discussed, it can be removed.
The description of apparatus and used techniques are too wordy and can be reduced due to the removal of unnecessary details.
Table 2 is too big and provides data that is not easy to read and analyse: It unclear: where is Figure 8 pictures (b0-b4)? It probably should be Fig.5?, but What about pictures a0-a-4?)? Which positions do the authors mean in Table 2 (I suppose there are too many numbers on pictures belonging to different spectra)? It will probably be better to provide some graphs with changing patterns for the essential components instead.
What time do you mean in the capture of Figure 6 (The 0 ln( / ) w w with time in slag)? It is not understood what are the conditions of a process which timeline have you described here?
How to understand the statement in Line 169: …The content of FeO next to n·C2S-C3P solid melt was obtained in C2S, for the left FeO may play a role of eroding the unreacted C2S….?
It seems that lines showing the relationship between P2O5 and FeO in C2S in Figure 7 are drawn too voluntary. It is necessary to show at least the determination coefficients (R2).
The schematic diagram (Figure 8) does not show any reactions going in the process between C2S, and molten slag is too easy and, therefore, is not informative and can be removed to save place.
The surface picture of the solid solution layer looks very beautiful but the appearance of “…shedded small particles showed a bulged and hollowed shape..” is not very visible as well as the conclusion about bigger and smaller content of phosphorus looks is not convincing taking into consideration the slight difference in found contents and particle sizes. The authors confirm this by themselves in the following sentence.
And again, in lines 212-213 - it is not the right way to explain one word using the same words ” inhomogeneous due to composition inhomgeneity”: …The residual stress was made 212 in a solid particle when density was locally inhomogeneous due to composition inhomogeneity.
In Figure 11 the numbers on the ordinate axle look were scaled voluntary because the intensity for other compounds than BaO should not grow together with BaO content increasing.
The location of points 1-4 in Figure 14 is not shown, and the result of the area scan analysis showing the mapping of the content of elements does not fully confirm the conclusion made by the authors.
I suppose the article should be thoroughly reread and slightly rewritten by authors: shorteneng methods descriptions in all parts and adding more facts and explanations confirming the mechanism of found phenomena.
Author Response

(The authors gave the same response as above.)
